# Structural Basis of Activity of HER2-Targeting Construct Composed of DARPin G3 and Albumin-Binding Domains

**DOI:** 10.3390/ijms252111370

**Published:** 2024-10-22

**Authors:** Anastasia G. Konshina, Eduard V. Bocharov, Elena V. Konovalova, Alexey A. Schulga, Vladimir Tolmachev, Sergey M. Deyev, Roman G. Efremov

**Affiliations:** 1Shemyakin & Ovchinnikov KT, Russian Academy of Sciences, Moscow 117997, Russia; konnij@gmail.com (A.G.K.); edvbon@mail.ru (E.V.B.); elena.ko.mail@gmail.com (E.V.K.); schulga@gmail.com (A.A.S.); biomem@mail.ru (S.M.D.); 2Moscow Center for Advanced Studies, Moscow 123592, Russia; 3Research Centrum for Oncotheranostics, Research School of Chemistry and Applied Biomedical Sciences, Tomsk Polytechnic University, Tomsk 634050, Russia; 4Department of Immunology, Genetics and Pathology, Uppsala University, 751 85 Uppsala, Sweden; vladimir.tolmachev@igp.uu.se; 5Laboratory of Molecular Pharmacology, Institute of Molecular Theranostics, Sechenov First Moscow State Medical University (Sechenov University), Moscow 119991, Russia; 6Department of Applied Mathematics, National Research University Higher School of Economics, Moscow 101000, Russia

**Keywords:** scaffold proteins, HER2, tumor targeting, NMR, molecular modeling, protein-protein interactions

## Abstract

Non-immunoglobulin-based scaffold proteins (SPs) represent one of the key therapeutic target-specific and high-affinity binders in modern medicine. Among their cellular targets are signaling receptors, in particular, receptor tyrosine kinases, whose dysfunction leads to the development of cancer and other serious diseases. Successful applications of SPs have been reported for HER receptor type 2 (HER2), a member of the human epidermal growth factor receptor family that regulates cell growth and differentiation. To extend the blood residence of SPs and prevent their high accumulation in the kidneys, these proteins are often fused with serum albumin. Promising results for HER2-binding activity were obtained for SP G3 from the DARPins (Designed Ankyrin Repeat Proteins) family fused with an albumin-binding domain (ABD). Interestingly, the detected HER2–G3 binding strongly depended on the position of the G3 module in the sequence of the constructs. Further improvement of these constructs for biomedical applications requires deciphering the molecular mechanism responsible for this effect. Here, we investigate the structural and dynamic aspects of ABD–G3 and G3–ABD chimeras using NMR spectroscopy and molecular modeling. Based on biophysical data, we come to the conclusion that extensive inter-domain contacts form in both constructs, although their binding interfaces and complex stability are somewhat different. Also, it is shown that the domain linker plays an important role—it limits the accessibility of the detected protein–protein binding sites, depending on the order of the domains in the chimeric molecules. These results create a solid structural basis for the rational design of new effective SP constructs targeting the signaling receptors in cells.

## 1. Introduction

Disorders in the functioning of cellular signaling receptors, in particular, members of the receptor tyrosine kinase (RTK) family, lead to the development of a number of serious diseases, primarily various types of cancer. One of the effective therapeutic strategies in this case is the use of molecules that selectively bind to the malfunctioning receptor and modulate its activity in a specified manner. Besides monoclonal immunoglobulin-based antibodies (mAbs), which are one of the key therapeutic target-specific and high-affinity binders in modern medicine [1,2], non-immunoglobulin-based scaffold proteins (SPs) are also widely used for the generation of novel specific binding agents [3]. A robust structural scaffold of SPs allows for the effective reshaping of their interaction sites for the selective recognition their cellular molecular targets. Compared to mAbs, SPs display a number of advantages, such as a significantly smaller (up to an order of magnitude) size and a high thermal stability, which result in relative simplicity and low-cost production in bacteria [4]. Some SPs (Adnectins, Affibodies, Anticalins, DARPins, etc.) have already attracted the close attention of the pharmaceutical industry, entering the clinical trial phase [3,5,6]. Successful applications of SPs have been reported for some members of the human epidermal growth factor receptor (HER) family that regulate vital cell processes, including cell growth and differentiation [3,5,7]. These receptors consist of a large extracellular ligand-binding domain, a single transmembrane helix, and a cytoplasmic domain with tyrosine kinase catalytic activity. Among them, one of the most important representatives is HER receptor type 2 (HER2). Its overexpression, which occurs in a number of cancers, including the most common breast cancer, is often associated with a high rate of cell proliferation and more aggressive disease leading to multiple distant metastases [8,9]. The arsenal of HER2-targeting therapy includes substances (first of all, mAbs and SPs) that selectively bind to various epitopes of the HER2 molecule and specifically modulate its activity [7,10]. Preclinical studies have demonstrated that SPs can be successfully applied in radionuclide molecular imaging for detecting tumor metastases, as well as for controlling the efficacy and dynamics of chemotherapy for HER2-expressing cancers. The evaluated SP-based probes were shown to accumulate in tumors with a high efficacy and help to elicit HER2-expressing cancers with a high imaging contrast, being well-tolerated and quite safe [11,12,13].

Promising results for the future development of SP-based probes for antitumor HER2-targeting therapy were obtained in a recent study of the HER2-binding activity of SP G3, which belongs to the DARPins (Designed Ankyrin Repeat Proteins) family [14]. The relative simplicity and rapid generation of libraries of highly specific DARPins with a subnanomolar affinity for a variety of target molecules, including various kinases, make this protein scaffold a good candidate for drugs [15,16]. Due to their physico-chemical properties, DARPins are widely used in basic research, for example, as biosensors that track the localization of proteins and interactions within cells [17,18]. Like other repeat proteins, the DARPin molecule is composed of repeating conserved structural motifs that mimic the tandem structure of natural Ankyrins found in many proteins involved in specific protein–protein interactions and that provide cell signaling and regulation, cytoskeleton integrity, inflammatory response, and many other functions [19].

However, one of the key problems limiting extensive clinical trials of SP-based therapeutic agents, such as the G3 protein, is their short residence time in the bloodstream due to efficient kidney filtration [20]. A number of strategies to extend blood residence and prevent the high accumulation of small therapeutic proteins in the kidneys have been proposed [21]. Today, the dominant approaches follow the path of increasing the size of biotherapeutics and their hydrodynamic radius, thus overcoming the renal filtration cutoff (ca. 60 kDa). An alternative strategy for extending half-life is fusion with the Fc region of IgG or serum albumin for recycling by the neonatal Fc receptor. Albumin is the most abundant transport protein in human serum with a very long half-life (ca. 19 days) and is, therefore, well-suited for experiments to increase the residence time of a pharmaceutical substance in the bloodstream [22]. The attachment of albumin-binding domains/peptides or the direct fusion of albumin with a small protein binder is believed to be a promising approach to further extending half-life due to the overlap between these two strategies [21,23,24]. Among a variety of fused albumin-binding moieties, both the designed serum albumin-binding domain (ABD) of DARPin [25] and the bacterial ABD-domain [14,24] can seriously increase the half-life in blood plasma, reaching the level of serum albumin. It was shown that a recently designed fusion construct, consisting of G3 and ABD modules, effectively modified the activity of HER2 [14]. To develop radionuclide therapeutic agent with an extended residence in the blood, reduced renal uptake, and a higher accumulation degree in the tumor, the authors created and tested, in vitro and in vivo, two fusion proteins (labeled with ^177^Lu) containing G3 and ABD domains connected by a (GS_3_)_3_-linker. It should be noted that the chimeras differed from each other only in the order of their domains. The variant with the ABD domain at the N-terminus demonstrated a loss of HER2-binding activity (but only in albumin-free medium). Nevertheless, both protein constructs were shown to have an increased plasma lifetime compared to the G3 protein used as a control.

The molecular mechanism responsible for these unexpected effects has not been established. It should be noted that little is known about the spatial rearrangement of the modules in the absence of their target partners. Both the fact of domain interaction and/or “inappropriate” binding interfaces can seriously affect the functionality of the entire protein construct. This means that the future development of such modular proteins, specifically those interacting with HER2 (as well as with other receptors and not only those from the RTK family) and proteins performing auxiliary functions (for example, albumin, which improves the pharmacokinetic profile of the chimera), requires understanding the molecular details of the chimera’s behavior in vivo. Furthermore, in modular proteins, the linker length and flexibility obviously affect their kinetic behavior and effective functioning [26], and, hence, should also be taken into consideration. To proceed with this, in the present work, we explore the molecular mechanisms underlying the differences in the target specificity of the ABD–G3 and G3–ABD constructs described in [14].

## 2. Results

### 2.1. Design of the Study

Obviously, the development, optimization, and practical application of the designed specific HER2 binder (G3 DARPin domain) require the elucidation of the molecular details of the structural mechanism of their interaction with the target receptor. As shown previously [14], the fused serum albumin-binding (ABD) domain extends the plasma life-time of both conjugates, G3–ABD and ABD–G3. To minimize steric hindrance between domains, a sufficiently long and flexible (GS_3_)_3_ linker between domains was added (Figure 1). However, varying the order of the domains was demonstrated to change the resulting activity. To understand the possible reason for this, a comprehensive multi-stage molecular biophysical approach was applied, including NMR spectroscopy and molecular modeling. The simulations included homology modeling, molecular docking, and atomistic molecular dynamics calculations. The objectives were formulated as follows:To establish whether the G3 and ABD domains in the conjugates are able to associate in an aqueous solution.If so, what are the most likely interfaces in the resulting complexes? Can they affect the known G3 binding site for the HER2 receptor?How can the effect of HSA be explained on the basis of molecular structure data?What determines the significant difference in the action of chimeras with the “opposite orientation”—G3–ABD/ABD–G3?

In NMR studies, high-resolution ^1^H-NMR spectra were obtained in water for both chimeras and mixtures of isolated fragments of G3 and ABD—with and without the addition of HSA. As a result, conclusions were made about the possibility of G3–ABD association, the general features of the conformational dynamics of proteins, and concerning the influence of HSA. In computational experiments, the propensity of the domains for association was evaluated using molecular dynamics (MD) and Monte Carlo (MC) methods, and possible protein–protein interactions were characterized in detail. Based on the totality of the biophysical data, we hypothesize about the possible role of the domain linker, which limits the accessibility of detected sites, depending on the order of the domains in chimeric molecules.

We would like to emphasize that the establishment of atomic-resolution NMR models is beyond the scope of this work. The main reason for this lies not in technical difficulties, but in the project objective. Thus, from a fundamental point of view, the laborious definition of such structures has no special value at this stage of the research project, since we are talking about not yet optimized artificial protein hybrid structures designed to solve specific practical problems of regulated action on the activity of the HER2 receptor [14]. For the further optimization and effective use of these constructs in biomedical applications, the observed effects have to be explained on a molecular basis, without detailed analyses of 3D structures. Therefore, the aforementioned combined “NMR + modeling” variant of analysis for various scenarios of complexation of the G3–ABD/linker was chosen.

### 2.2. NMR Spectroscopy

According to the characteristic signal dispersion in the ^1^H-NMR spectra of the DARPins G3–ABD and ABD–G3 (Figure 2 and Appendix A), both constructs have a globular structure consisting of α-helices, which were quite stable at 30 °C, without aggregation and denaturation, for at least a week. Nevertheless, the comparison of NMR spectra revealed some differences in the overall conformation of the two constructs, associated with the alternative spatial arrangement and interaction of their G3 and ABD domains. In particular, the distribution pattern of the ^1^H-signal differed markedly in the regions of the amide (Figure 2A) and methyl (Figure 2B) groups of the spectra. In both cases, the methyl signals of the Ile and Leu residues of the G3 domain were strongly shifted when G3 was linked to ABD (Appendix A). Remarkably, in the ^1^H-NMR spectra of ABD–G3, a high-field ^1^H-signal near −0.2 ppm (Figure 2B and Appendix A) of the methyl group of one Ile residue (spatially adjacent to the aromatic ring of Phe, according to the two-dimensional ^1^H-NMR spectra of G3, Appendix A) from the G3 domain was strongly split into two signals, with an occupancy ratio of the major and minor components equal to ~3. A similar distinct splitting was observed for a separate low-field amide signal of the G3 subunit near 9.6 ppm (Figure 2A), indicating that two distinct conformations of the ABD–G3 chimera existed in solution.

In the ^1^H-NMR spectra of G3–ABD, almost all protein signals also appeared to be doubled, but very broadened (especially those of the minor component), implying that conformation exchange, presumably between multiple interaction modes of subunits, occurred on an intermediate NMR timescale (in microseconds) for the G3–ABD variant, while the ABD–G3 construct was more rigid, with better mutual folding of domains via two alternative interfaces switching at a slow NMR timescale (in milliseconds and slower). It should be also noted that the free G3 and ABD interacted via non-covalent association in a fashion that was somewhat similar to the conjugated G3–ABD, and the complex was destroyed after the addition of HSA (see the Appendix A).

### 2.3. Molecular Modeling

According to the results of the MD and MC simulations, starting from both spatially remote domains and complexes with a hydrophobic interface (see Section 4), the G3 and ABD modules interacted with each other. It is important to note that, in the case of applying both computational approaches, the spatial structure of each of the individual domains remained stable (average root-mean-square deviation (RMSD) values of the backbone atoms from the starting states do not exceed 1.5 Å, see Appendix A), unlike the case of the flexible linkers between them. The analysis of the ensemble of all obtained complexes showed that the G3 module interacted with ABD through two distinct types of sites spatially distant from each other. In one case, the protein–protein interface was formed by hydrophobic residues of G3 (hereinafter, this nonpolar site is called “np-site”), and in the other by predominantly polar residues (“p-site”).

Detailed consideration of both G3 sites shows the following.

#### 2.3.1. The HER2- and ABD-Binding Interfaces (np-Site) on the G3 Surface Are Almost Identical

Domain packing via the np-site was quite stable, since the MD/MC simulations starting from both chimeras associated via this interface showed an absence of the dissociation of this complex. In these cases, the G3–ABD and ABD–G3 conjugates showed a high degree of similarity between the binding motif of G3 with the HER2-binding site in the experimentally obtained complexes of G3 with extracellular domains of membrane-bound HER2 receptors (PDB: 4hrn, Figure 3A) (Figure 3B,C). The identity of the contact areas in the predicted complexes and the experimental models was about 90%. The HER2-binding interface of G3 included a major nonpolar pattern on its molecular surface. It contained about 20 amino acid residues, most of which were hydrophobic. Interface-forming residues of G3 were distant from each other in the sequence and belonged to both ordered elements (α-helices) and β-hairpin/loop regions. Additional details of the predicted complexes are given in the Appendix A.

#### 2.3.2. Alternative p-Sites of ABD Binding on G3 Surface Were Found in Both Chimeras and Practically Do Not Overlap

The alternative interaction interfaces (p-sites) were characterized by a significantly smaller contact area with an absence of stacking interactions and fewer hydrophobic contacts. At the same time, the electrostatic interactions were more pronounced (Appendix A). In Figure 3, the most populated spatial locations (p-sites) of the ABD domain in the G3–ABD (Figure 3B) and ABD–G3 (Figure 3C) chimeras are shown in a light blue color and cartoon mode. In this figure, G3 surfaces with a high and low frequency of involvement of the corresponding residue in contact with the ABD molecule are colored in red and white. In nine 200 ns MD trajectories, where the domains were separated from each other (hereafter, these MD starts are designated as “w-MD starts”), only in ~20% of MD states the domains did not interact. Moreover, there were no MD trajectories where the domains remained non-interacting during the entire MD calculation time. During w-MD, the domains were usually associated during the first 100 ns, which is not surprising, since the molecular surface of G3 has a high net negative charge (−10). It seems likely that such p-sites were the result of the random sticking of the ABD domain to the G3 domain. It should be noted that the length and location of the inter-domain linker relative to the G3 domain are very important because they limit the area of the G3 surface accessible for such ABD “walking”. As can be seen from Figure 3B,C, potential ABD binding p-sites occupy large areas, but they practically do not overlap for two chimeras.

#### 2.3.3. The Availability of HER2 Site Depends on the Order of the Domains in the Constructs

In all w-MD runs of ABD–G3, there was a pool of HER2 interface residues that were part of the p-sites in most of the trajectories, as follows: Y46, L48, D77, A78, I79, and F112 (Figure 3B, hereinafter, the numbering of the residues corresponds to the G3–ABD sequence). In the case of ABD–G3, a “polar” variant of the domain interaction area considerably overlapped with the np-site, meaning that the ABD domain can sterically interfere with G3 binding to HER2 (Figure 3A,C).

On the contrary, for the G3–ABD molecule, all four w-MD trajectories showed “sticking” of ABD outside the np-site zone (Figure 3B). Such a difference was determined by the order of domains in the amino acid sequences of the chimeras and by the limited linker length.

As expected, the predicted ΔG values for the experimentally obtained HER2–G3 complex were lower (−7.5/−6.5 kcal/mol for AD/BC subunits) than the average ΔG values for the MD states of the chimeras with an np-interface (−5.5/−4.5 ± 0.5 kcal/mol for G3–ABD/ABD–G3) and especially p-interfaces (in the range from −3.6–−4.6 kcal/mol for the top clusters, see Appendix A). In the case of the np-interface, the less pronounced difference in ΔG values cannot exclude competition between the HER2 receptor and the ABD domain for the HER2 site of G3.

As demonstrated in [14], the addition of HSA into medium restores the activity (HER2 binding) of the ABD–G3 construct. It can be assumed that HSA, unlike HER2, more efficiently displaces chimera domains upon contact. Indeed, the highest binding affinity was predicted for the experimental model of ABD-HSA (−8.1 kcal/mol).

#### 2.3.4. The Involvement of the HSA Motif in G3 Binding Depends on Both the Type of Construct and the Binding Interface of Its Modules

Our simulation data show that, in both chimeras, the HSA motif of the ABD domain is involved in binding to G3. The HSA-binding motif elucidated from the high-resolution structures of ABD/HSA complexes (PDB ID: 2vdb) includes residues of two helices (except for the N-terminal one) and two short inter-helical loop regions which are spatially separated (Figure 4A,B and Appendix A). One of these loops is involved in H-bonding (Figure 4B) with the albumin molecule, while the other participates in charge interactions (Figure 4A). Between the loops, there is a hydrophobic cluster formed by helical residues. Different types of intermolecular interactions (H-bonds, electrostatic, and hydrophobic ones) provide the strong binding of the ABD domain to the albumin molecule.

In MD complexes with np-sites of G3, the HSA motif is almost completely inaccessible to the solvent (ca. 70% of HSA-binding residues are on the G3–ABD interface, see Appendix A) in both chimeras, and the binding of HSA to such complexes is impossible. However, in complexes with p-sites of G3, this motif is shielded from water to a different extent, as it is largely inaccessible to the solvent in ABD–G3 (Figure 4B), but rather exposed in the G3–ABD (Figure 4A, Appendix A). So, in the most populated MD cluster (from one of w-MD runs) for ABD–G3, all three regions (two loops and helical residues) of the HSA motif interact with G3, resulting in more than 50% of the HSA interface being inaccessible (Appendix A). Obviously, in this case (as in the case of the np-interface for both complexes), albumin will compete with G3 for binding to ABD.

In contrast, the C-terminal location of the ABD domain results in a less extended area of the protein–protein interface, which is characterized mainly by electrostatic interactions arising from the inter-helical “charged” loop of the HSA motif carrying the corresponding residues (Figure 4A). Thus, in all w-MD trajectories of G3–ABD, a large part (on average, more than 70%) of the HSA site remains accessible in solvent. It can be assumed that, in this case, albumin can bind to the ABD domain, even within the G3–ABD complex, without its dissociation (Appendix A). However, it is quite possible that bound HSA may sterically interfere with the effective interaction of the G3–ABD complex with the HER2 receptor (as shown in the model reconstruction of the HER2–G3/ABD–HSA complex, Appendix A).

#### 2.3.5. Role of the Domain Linker

Analysis of the MD data revealed that the domain linker (12 residues) connecting the G3–ABD domains affected the spatial packing of the domains. So, the number of atom–atom contacts of the linker with both domains was ~50% of the domain–domain contacts in the G3–ABD complex with a hydrophobic interface (np-site). In the case of polar interfaces (p-sites), the number of such contacts turned out to be comparable to that observed for the inter-domain interactions (see Appendix A). Analysis of w-MD starts showed that, in the most populated states, the linker tended to interact with neighboring helical and flexible long loop regions of G3, which were spatially distant from each other in G3–ABD and ABD–G3 (Figure 5A,B). As a result, such linker behavior delimited the regions of the G3 surface available for ABD binding and may restrict the accessibility of the HER2 site depending on the chimera type.

Also, the domain linker can participate in the additional stabilization of the conjugates’ complexes due to their interaction with the domains (mostly with G3). Thus, in the case of ABD–G3 complexes with a p-interface, the average contact area of the linker with G3 was around two times greater than that in the analogous G3–ABD complexes, and the complex was additionally stabilized by a larger number of H-bonds between the linker and G3 (see Appendix A). Taking the linker into consideration and conducting a subsequent recalculation of the binding affinity (in terms of ΔG values) between G3 and the domain containing the ABD and the linker (see Figure 1) resulted in an increase in the predicted affinity for both types of complexes (see Appendix A). Moreover, in case of MD complexes with polar interfaces, we can see that the ΔG values decreased for both constructs, but more significantly for the C-terminal G3 construct (−6.6 vs. −4.7 kcal/mol for the most populated clusters of ABD–G3 and G3–ABD, respectively).

## 3. Discussion

As was shown in vitro [14], the binding of ABD–G3 to human serum albumin (HSA) was substantially weaker compared to that of G3–ABD. Moreover, without albumin the specific binding of ABD–G3 to HER2 was lost. Here, we explain these results at the molecular level based on NMR and modeling data. As revealed by the NMR spectra, G3 and ABD mutually associated both in conjugates and in free states via at least two distinct alternative interfaces with an occupancy ratio of ~3:1. This agrees perfectly well with the results of the modeling, where both constructs revealed a high potential for G3–ABD interaction—two types of binding interfaces (the so-called np- and p–sites) were identified in the G3 domain. The hydrophobic contact region of G3 (np-site) was validated by NMR-data, showing that apolar side chains of Tyr, Leu, and Ile residues (spatially adjacent to one of phenylalanines) from G3 participated directly in G3–ABD interaction. Indeed, two pairs of Ile/Phe (I79/F81 and I79/F112), as well as a number of Leu (L48, L53, and L86) and Y46 residues, were found in the np-site observed in simulations and in the experimentally derived HER2-binding site. It is important that, in complexes with such a hydrophobic interface, the contact regions in the G3 and ABD domains are almost identical to the known HER2- and HSA-binding sites, respectively. Compared to the hydrophobic np-site, the spontaneously formed domain complexes observed in the MD/MC simulations starting from non-interacting protein domains had much smaller and more polar contact areas (p-sites). It should be noted that alternative inter-domain interfaces for both conjugates were also detected by NMR.

Based on the experimental and modeling data, we propose the following scenario explaining the detected changes in the in vitro activity of ABD-fused conjugates, depending on the order of their domains. The main findings are summarized and presented schematically in the diagram below (Figure 6). Depending on the chimera, the alternative ABD binding sites (p-sites) on the G3 surface had significantly different locations (Figure 6A). The computational results prove that the inter-domain linker delimited the regions of the G3 surface (Figure 3) available for ABD binding, depending on its C- or N-terminal location in the sequence, and affected the accessibility of the HER2 site. Importantly, the p-sites in G3–ABD did not overlap with the HER2/np-site. Thus, the HER2-binding site in G3–ABD remained free and capable of interacting with the receptor. The opposite situation was observed in ABD–G3, where the region of overlapping interfaces (np- and p-sites) included Ile and Phe residues (found on the interfaces in all simulations). Taking the linker into consideration and conducting a subsequent recalculation of the binding affinity (in terms of ΔG values) between G3 and the domain containing the ABD and the linker resulted in an increase in the predicted affinity for both types of complexes (see Appendix A). Thus, regardless of the occupancies of the G3–ABD binding interfaces, ABD–G3 is likely to be sterically constrained to interact with HER2 and lose its activity in an albumin-free medium (Figure 6A).

Since both chimeras contained the ABD modulewhich has a high-affinity for albumin, the letter added to the medium strongly competed for interaction with G3. This resulted in dissociation of the G3–ABD complexes (Figure 6B), exposure of the HER2/np-site, and the restored binding of ABD–G3 to HER2 (Figure 6C).

For complexes with polar interfaces, the predicted ΔG values were even higher than for ones with an np-site. The detected polar sites were multiple and clearly not specific. So, such complexes can be considered as transit states in which the complex can “get stuck”. At the same time, if the linker–G3 interaction is taken into consideration, the recalculated ΔG values for complexes with polar sites become lower and comparable with np-interfaced complexes (Appendix A). For example, for the MD states of ABD–G3, in which the ABD domain partially interacted with the HER2 site (thus “preventing” the HER2 binding, see Figure 3B and Figure 6A) of the G3 domain, the average ΔG values calculated for G3–ABD binding and the G3–ABD + linker were −4.0 ± 0.3 and −6.6 ± 0.7 kcal/mol, respectively.

The differences in the dissociation rates of the complexes with p- and np-interfaces could have played a significant role in reducing the affinity of ABD–G3 for HSA observed in the experiments. As can be seen in the diagram (Figure 6A), the direct binding of albumin to ABD–G3 was not possible due to the total (in the case of the np-site) or substantial (p-sites) inaccessibility of the HSA site in the ABD domain. Moreover, as revealed by NMR, the domains in ABD–G3 were better packed than those in G3–ABD and revealed slower (milliseconds and slower) transitions between alternative association modes. Taking into account the modeling data, we also suggest that the domain linker contributed to the additional stabilization of the complexes with p-interfaces. In contrast to ABD–G3, the energy barrier between the major and minor conformational states of G3–ABD was substantially less, resulting in fast switching between these interaction modes (on a microsecond NMR timescale). As a result, both the HSA and HER2 sites were more exposed and available for interaction with albumin and the HER2 receptor. In addition, based on the significantly lower involvement of the HSA motif in minor states of the G3–ABD complexes (with p-interfaces), it can be assumed that the interaction of this complex with both targets (HSA and HER2) can occur without the dissociation of the complex (Figure 6B).

We also found that the targeting properties of the ABD-fused construct were sensitive to a number of structural and dynamic factors, which, in turn, depended on the order of the domains in the molecule. These included the location of the G3–ABD interaction interfaces, the overlap between different types of interfaces, the ratio of the populations of complexes with alternative interfaces, and the degree of involvement of the HSA motif in domain interactions. So, a change in the order of the domains in the polypeptide chain can significantly affect the spatial packing of domains, shifting the equilibrium distribution of possible types of ABD–G3 complexes, which leads to changes in their biological activity.

In summary, this study highlights that an unstructured flexible linker can promote undesired associations between functional domains. The degree of involvement of functionally important binding sites of the domains in such unwanted domain dimerization depends strongly on the order of these domains. Taking these factors into account, the following directions for the future modification of domain linkers in the designed constructs can be proposed: adjusting the linker length by making it shorter and/or using a stiff a-helical linker to keep a fixed distance between domains, and preventing domain interactions. We believe that the proposed approach, combining NMR measurements and computer modeling techniques, will allow researchers to quickly assess the effectiveness of amino acid sequence changes at the structural level and, thus, more rationally plan labor-intensive in vivo experiments.

Finally, it is important to note that the aim of this work was not to determine the exact (with a high resolution) spatial structure(s) of the complexes of G3 and ABD domains constituting the chimeras. The solution to this problem lies beyond the scope of this study—the laborious definition of such structures has no special value at this stage, since we are talking about not yet optimized artificial protein hybrid structures designed to solve specific practical problems of regulated action on the activity of the HER2 receptor [14]. For the further optimization and effective use of these constructs in biomedical applications, the observed effects have to be explained on a molecular basis, without detailed analysis of 3D structures. These findings will prove useful in guiding ongoing experiments aimed at improving G3–ABD chimeras. The predictive power of the proposed approach is confirmed both by the good agreement of several independent biophysical methods and their ability to explain the effects observed in the experiments. This approach is universal—in addition to the considered designs, it can be applied to other SP-based hybrid protein chimeras.

## 4. Materials and Methods

The amino acid sequences of G3, G3–ABD, and ABD–G3 conjugates are given in the Appendix A (Figure 1). The expression, isolation, and purification of the proteins were performed according to the methodology described earlier [14]. Most of the chemicals used in the study were purchased from Sigma-Aldrich, Sweden AB (Stockholm, Sweden).

### 4.1. NMR Studies

The ^1^H-NMR spectra were acquired at 30 °C on a 600 MHz AVANCE III spectrometer (Bruker BioSpin, Rheinstetten, Germany) equipped with a 5 mm pulsed-field gradient triple-resonance cryoprobe. NMR samples of the DARPins G3–ABD (120 µM) and ABD–G3 (140 µM), as well as the equimolar mixture of separated G3 and ABD (150 µM), were prepared in 5 mm shigemi NMR tubes using 20 mM phosphate buffer with pH 7.4 containing 200 mM NaCl and 10% D_2_O. Then, 1 mM tris (2-carboxyethyl) phosphine (TCEP) was used as a reducing agent to break the inter-molecular S-S bridge between the C-terminal residues of G3–ABD and ABD–G3.

### 4.2. Modeling of Interaction Between DARPin G3 and ABD

#### 4.2.1. Building Up the Starting Spatial Models of G3–ABD and ABD–G3

The spatial structure of the DARPin module (residues: 12–136) was taken from the Protein Data Bank (PDB ID: 2jab). Hereinafter, the numbering of the domains’ residues is given for the G3–ABD molecule. An NMR model of the albumin-binding domain of protein G from *Streptococcus* sp. (PDB ID: 1gjs, residues: 20–65) was selected as a structural template for the ABD module (residues: 149–194). The sequences of the template and the ABD domain are very similar (>85% identity, only 7 amino acid residue replacements, see Appendix A). The homology model of ABD was built using the Modeller software (version 9.9) [27]. The final spatial structures of the entire G3–ABD and ABD–G3 molecules (200 and 196 amino acids, respectively, Figure 1), including a 12-mer polypeptide linker between the G3 and ABD modules, were constructed with Modeller (version 9.9) and Maestro (version 9.3.5) software.

To assess the interaction potential of the DARPin (G3)–ABD modules, all-atom MD simulations were carried out in an explicit water solution. The following variants of starting G3 and ABD mutual arrangements in both the G3–ABD and ABD–G3 molecules were prepared:Domains are distant from each other and do not interact;Domains are in contact and both have a “hydrophobic” binding interface.

The starting model of this complex was obtained using a protein–protein docking procedure and ZDOCK version 3.0.2 program (https://zdock.wenglab.org/ accessed on 16 September 2024). Before, the distribution of the hydrophobic/hydrophilic properties on the molecular surfaces of the corresponding protein modules was calculated using the molecular hydrophobicity potential (MHP) approach implemented in the PLATINUM version 1.0 software [28]. Then, some hydrophobic residues (G3: F89, I90; ABD: F168, Y169) from the central part of the major hydrophobic patterns were selected as “contacting” for the rescoring of the top docking solutions in such a way that the specified residues were in the binding site. Among the top 10 selected poses (provided that the “contacting” residues were at the interface), we selected the pose with the best score value.

#### 4.2.2. MD Simulations

A number of 200 ns MD simulations in a water solution were carried out for the G3–ABD and ABD–G3 proteins. MD calculations were performed using the GROMACS package (versions 2020.4/6) [29] and the all-atom force field CHARMM36 [30]. In all calculations, the tip3p [31] water model and 3D periodic boundary conditions were employed. A spherical cutoff function (12 Å) and the particle mesh Ewald (PME) algorithm [32] (with a 12 Å cutoff) were used to treat van der Waals and electrostatic interactions, respectively. The preparation stages included system minimization and heating from 5 to 310 K. Finally, MD production runs (durations of 200 ns each) were conducted in an NPT ensemble at a constant temperature of 310 K with an integration step of 2 fs. A brief description of the starting systems and the number of MD trajectories is given in Appendix A.

#### 4.2.3. MC Conformational Search

The protein conformational space was explored via MC search in torsion angles space and in the presence of an implicit water-mimicking solvent, as described elsewhere [33]. Two starting models of the protein complex G3–ABD were employed, as follows: (i) included as randomly placed non-interacting domains and (ii) associated domains with the hydrophobic binding interface (initial as the MD state taken (from MD runs) of the associated domains with a hydrophobic binding interface). MC simulations were performed using the all-atom ECEPP/2 force field [34]. The distance-dependent dielectric permeability ɛ = 4 × *r* and an adaptive-temperature schedule protocol were applied. After each MC step, the structures were subjected to conjugate gradient energy minimization. Other details of the MC simulation protocol are given in [33]. Further analysis of the obtained G3–ABD binding modes was carried out for the resulting ensemble of low-energy MC states in the energy range E_min._, E_min._+ 20 kcal/mol, where E_min_ is the minimal potential energy of the system obtained via MC simulations.

#### 4.2.4. Data Analysis

Analysis of the MD/MC data included a detailed analysis of the interaction interface between the DARPin and ABD modules in G3–ABD and ABD–G3 molecules. MD data were analyzed and averaged over the resulting MD trajectories. The latter were sampled at time intervals of 1000 ps. Conformational mobility and intermolecular contacts (including hydrophobic, electrostatic, and stacking interactions) were delineated using GROMACS [29] tools and original in-house utilities. To identify the distinct binding modes (np- and p-sites) between the G3 and ABD domains, MD data were clustered using the RMSD-based clustering algorithm (gromos) implemented in the GROMACS software with a 2Å cutoff for backbone atoms. To compare the binding affinities (ΔG) of experimentally derived HER2–G3/HSA–ABD complexes with simulated G3–ABD ones, structurally based predictions of ΔG values for the corresponding models from PDB:4hrn/2vdb and a set of MD states of chimeras that show the np- and the most stable p-interfaces were performed with the help of the web-server PRODIGY (https://rascar.science.uu.nl/prodigy/ accessed on 11 October 2024). This method is based on an analysis of the interfacial contacts of experimentally derived protein–protein complexes and the search for correlations with their experimental binding affinities. For both chimeras, in the case of MD runs starting from docking complexes or from distantly located domains, all MD states or the two most populated (in the corresponding trajectories) MD clusters, respectively, were chosen for ΔG calculation. To assess the contribution of the linker (G3–linker interactions), we calculate the binding affinity (designated as ΔG_L in Appendix A) between G3 and the domain containing ABD and the linker for the corresponding MD states of chimeras. To avoid the contribution of the linker residues covalently linked to the G3 domain, two adjacent linker residues were excluded from consideration. The distribution of hydrophobic/hydrophilic properties on the molecular surfaces of the peptides was calculated using the MHP approach [35]. Mapping the hydrophobic properties of the peptides’ surfaces was estimated with the PLATINUM software [28]. The accessible surface area (“contact area”) was calculated by the naccess version 2.1.1 program (http://www.bioinf.manchester.ac.uk/naccess/ accessed on 16 September 2024). Molecular graphics were rendered using PyMOL v. 2.5 (http://pymol.org accessed on 16 September 2024).

## 5. Conclusions

As shown earlier, the fusion of anti-HER2 DARPins with ABD increases their retention time in the blood and leads to a higher accumulation in the tumor due to the high bioavailability of the targeting agent. The main finding of this work is that the effect of merging SP with ABD depends strongly on the order of domains in the construct and is most likely caused by the domain–domain association and the different conformation/positioning of linkers with respect to these domains, which affects the domain–linker interactions. This suggests that the future design of antitumor conjugates based on DARPin–ABD fusion and related chimeras requires the careful evaluation of various protein architecture using molecular biophysics methods.

## Figures and Tables

**Figure 1 ijms-25-11370-f001:**
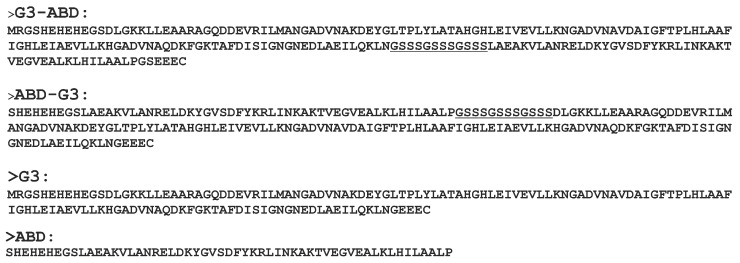
Amino acid sequences of DARPin constructions: G3–ABD, ABD–G3, G3, and ABD. The sequence of the interdomain linker (GS_3_)_3_ is underlined.

**Figure 2 ijms-25-11370-f002:**
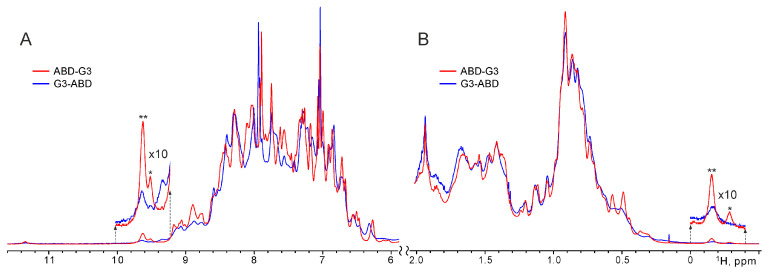
Overlaid ^1^H-NMR spectra of the DARPins G3–ABD (in blue) and ABD–G3 (in red). The characteristic regions with the signals of amide and aromatic groups (**A**) and methyl groups (**B**) are shown. (The full spectra are presented in Appendix A.) The separate low-field and high-field ^1^H-signals (near 9.6 and −0.2 ppm) of the G3 subunit are zoomed, and the major and minor components of the split ^1^H-signals are marked by asterisks.

**Figure 3 ijms-25-11370-f003:**
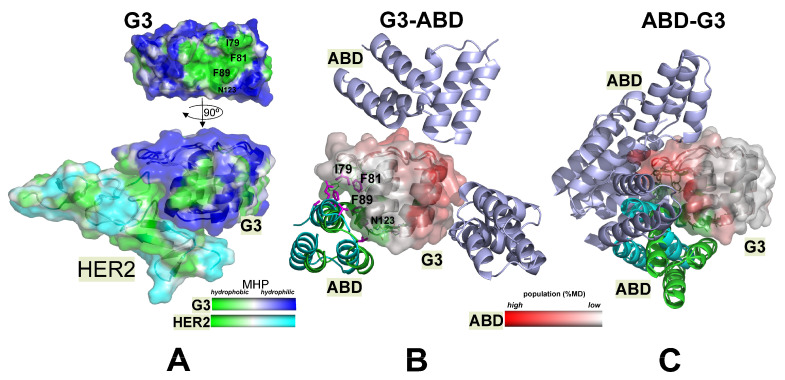
MD/MC-data analysis: difference in binding interfaces of the G3 domain for G3–ABD and ABD–G3 molecules. (**A**) The molecular surfaces of both DARPin (G3) domain and experimentally-derived model of the HER2 (subdomain IV)–G3 complex (4hrn, D) are colored according to the values of the molecular hydrophobicity potential (MHP). The main hydrophobic pattern on the DARPin surface is shown (**A**). The molecular surfaces of G3 module of G3–ABD (**B**) and ABD–G3 (**C**) are colored according to the high or low degree of involvement of the corresponding residues in contact with the ABD module during w-MD-runs (starts from non-interacting domains). Spatial arrangement of ABD domain relative to G3 in MD complexes of G3–ABD (**B**) and ABD–G3 (**C**) chimeras is given in cartoon representation. The corresponding representatives of the most populated MD-clusters are colored in green/cyan and light blue for hydrophobic interfaces (np-site) and a number of “polar” interfaces (p-sites), respectively. The key interface residues of G3 found in MD trajectories starting from complexes with hydrophobic interface (**B**), in low-energy MC states (colored in magenta, (**B**)), and in experimental 3D model of HER2–G3 (**A**) are indicated.

**Figure 4 ijms-25-11370-f004:**
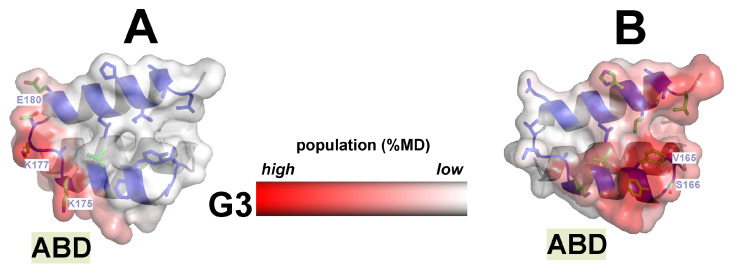
Difference in binding interfaces of the ABD domain for G3–ABD and ABD–G3 molecules. The molecular surfaces of ABD modules of G3–ABD (**A**) and ABD–G3 (**B**) are colored according to the high or low degree of involvement of the corresponding residues in contact with the G3 domain during w-MD runs. The corresponding scale is given with the spectral band. Data are averaged over all w-MD-trajectories, color intensity reflects the population degree. HSA-binding residues found in HSA–ABD complex (PDB ID: 2vdb) are represented by stick lines. The residues from inter-helical loops of HSA motif involved in electrostatic interactions (including H-bonds) with HSA are designated (see Appendix A). The HSA residues found at the G3–ABD interfaces of MD complexes obtained from MD simulations of non-interacting (at start) domains are colored by atom type.

**Figure 5 ijms-25-11370-f005:**
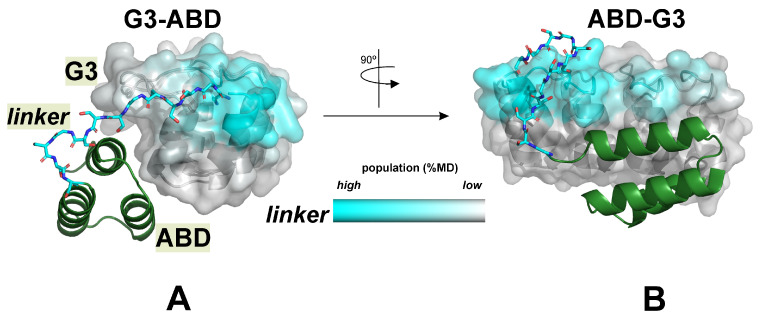
Results of MD simulations started from non-interacting G3 and ABD domains: difference in interaction interfaces of the domain linker for G3–ABD and ABD–G3 molecules. The molecular surface of G3 module of G3–ABD (**A**) and ABD–G3 (**B**) is colored according to the high or low degree of involvement of the corresponding residues in contact with the inter-domain linker during w-MD runs. Data are averaged over all w-MD trajectories, and color intensity reflects the population degree. Location of the linker (backbone atoms) in the complexes with the hydrophobic domain interface (ABD domain is given as green helix) is shown relative to the area of its most probable contact with the G3 surface observed in w-MD starts.

**Figure 6 ijms-25-11370-f006:**
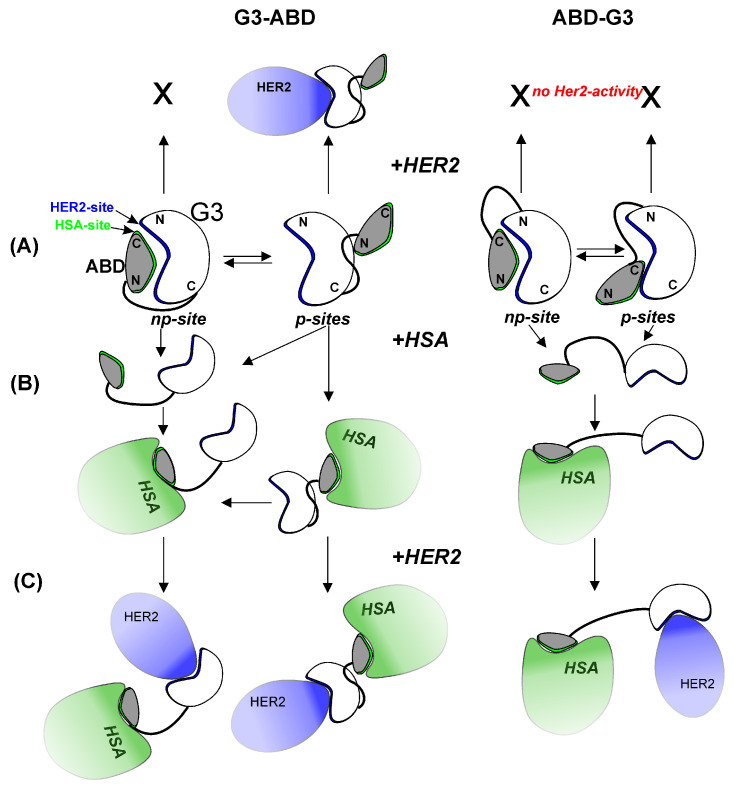
A hypothetical diagram illustrating the most likely protein–protein interactions predicted based on NMR and modeling experiments. (**A**) ABD-binding sites (p- and np-sites) on the surface of G3 domain of the chimeras G3-ABD and ABD-G3: p-sites have different localizations depending on the type of construct. Location of p-site near HER2-binding interface of G3 can lead to a loss of HER2-activity of the ABD-G3 in HSA-free environment. (**B**) Types of HSA/chimera complexes (with or without dissociation of G3/ABD domains) upon addition of HSA to the medium. (**C**) HER2-binding activity of the chimeras upon addition of HER2 to HSA-containing medium.

## Data Availability

The data presented in this study are available within the article and the Appendix A. Further inquiries can be directed to the corresponding author.

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
