# Peer review of "Structural Basis of Activity of HER2-Targeting Construct Composed of DARPin G3 and Albumin-Binding Domains"

_ijms, 2024, doi:10.3390/ijms252111370_

Round 1

Reviewer 1 Report

Comments and Suggestions for Authors

In this study Anastasia et al. report differences between the addition of an albumin binding domain to the N-terminal vs C-terminal of the G3 scaffold protein targeting HER-2. Due to the increased use of scaffold proteins as therapeutics, and the need to extend half-lives through the addition of tags, I think this is a timely and informative study highlighting major differences between adding it to the N or C terminus. Due to this, and the use of molecular simulations to understand conformations, this is an appropriate manuscript for this special edition of IJMS. However, I believe key experiments are missing that would support the conclusions made and complete the story. In addition, some sections are incredibly hard to follow and need major reworking.

My specific comments are below.

Major comments:

1)      I would expect to see more NMR data than just 1D experiments. For example, a 15N labelled HSQC would show differences in conformation and structure much more clearly and effectively. I do not expect a full backbone assignment (although that would be very powerful in this study) and take on board the authors point in the discussion that detailed structural determination was not the goal of this study. However, considering that NMR represents half the experimental figures, I think some 2D NMR data is needed.

2)      The fact the ABD and G3 domains interact with each other when not covalently linked to one another is glossed over too quickly and I do not find the 1D NMR that useful for showing this. I would like to see some simple experimental binding assays (e.g. SPR, ITC or ELISA) to show the affinities of these two domains for one another. This could also be used to more convincingly show that the presence of HSA inhibits this interaction.

3)       Figure 2 is incredibly hard to follow and needs to be redesigned if it is to be easily followed. This is not helped by the figure legend that does a rather messy job of guiding the reader through the images. It appears that many of the panels are interconnected which does not help. I would suggest more consistent labelling and colour coding of the different proteins. I also want to see clearer descriptions in the figure legend of what is being shown in each panel starting with which proteins are being shown to interact. The easiest way to achieve this is to split figure 2 into two or three separate figures.

4)      I struggle to see the interactions of G3 and ABD (and the linker) having a high affinity, especially compared with the affinities of G3 to HER2 and ABD to HSA. Therefore, surely the interaction of G3 and ABD will be easily displaced upon contact with HER2 or HSA. This begs the question, does it really make a difference that ABD-G3 has perturbed binding sites if they can be displaced easily? This is another area that some binding assays could help strengthen the study. In the presence of HSA, does ABD-G3 have a reduced binding affinity to HER1compared to G3-ABD also in the presence of HSA.

Minor comments:         

1)      Section 2.1, design of study. Would be useful to quote what the linker is in main text as well as showing in figure s1.

2)      Figure 1. The authors should highlight, with a marker or arrow, the peaks they are referring to in the text at -0.2 and 9.6 ppm as it is not that clear what peak splits are being mentioned.

3)      Figure s2C needs a bit more explanation as I am struggling see what it is showing. To my eyes, the 1D H-NMR spectra looks the same for the complex of ABD and G3 in both the presence and absence of HSA

4)      Figure 2 legend has the wrong PDB code (1HRN instead of 4HRN)

Comments on the Quality of English Language

The quality of the English is ok but needs to be carefully looked at and corrected as there are many minor mistakes throughout that make it harder to follow. 

Reviewer 2 Report

Comments and Suggestions for Authors

In the article “Structural basis of the activity of the HER2-targeting construct composed of DARPin G3 and albumin binding domains” the authors present a study to try to characterize the differences between an ABD-G3 fusion and a G3-ABD fusion. They present some NMR data and computational studies.

The logical sequence of the experimental and computational approach is fine, however, the execution lacks depth. It is a good preliminary study but it is not in a publication state.

Protein NMR is a very powerful technique that can provide several information. However, 1D protein spectra are only used as a measurement of the feasiblility of the proyect and have not been used as analysis spectra for several decades. There is just much overlap and does not provide results or information that more advanced experiments do. To really judge everything the authors claim, at least simple 2D hsqc spectra would need and to determine dynamics and exchange between states it would be require dispersion or other experiments. At the moment all NMR conclusions are just wishes.

As the authors realize, it is better to have experimental data to support the computational results.

The simulation would also benefit from a deeper analysis, where are the trajectory analyzes?. The entire analysis seems to be base on snapshot. The initial structures appear to be from docking. How was it done? How were the poses selected?

A docking program will always give interaction and a simulation strongly depend on the correct start and condition.

To evaluate the interaction and “intercompactness” of the complex experimentally, ITC, SAXS, DLS, etc. could be used.

The fact that obtaining reliable and analyzable results “would require a lot of effort both experimentally and computationally” it is not a justification for not doing them, nor a measure of the scientific relevance of the work.

Round 2

Reviewer 1 Report

Comments and Suggestions for Authors

I am happy with the changes made since the first submission and all my major comments have been largely addressed. I suggest the English is looked over again during final proofs.  

Comments on the Quality of English Language

Improved from before but some minor improvements can be made during proof reading. 

Reviewer 2 Report

Comments and Suggestions for Authors

The authors have responded to several of the points made.The paper have improve